# Blocking Linear Cryptanalysis Attacks Found on Cryptographic Algorithms Used on Internet of Thing Based on the Novel Approaches of Using Galois Field (GF (2$^{32}$)) and High Irreducible Polynomials

**Khumbelo Difference Muthavhine * and Mbuyu Sumbwanyambe *** 

School of Engineering, College of Science, Engineering & Technology, University of South Africa, Pretoria 0003, South Africa
* Correspondence: kdmuthavhine@gmail.com (K.D.M.); sumbwm@unisa.ac.za (M.S.)

**Abstract:** Attacks on the Internet of Things (IoT) are not highly considered during the design and implementation. The prioritization is making profits and supplying services to clients. Most cryptographic algorithms that are commonly used on the IoT are vulnerable to attacks such as linear, differential, differential–linear cryptanalysis attacks, and many more. In this study, we focus only on linear cryptanalysis attacks. Little has been achieved (by other researchers) to prevent or block linear cryptanalysis attacks on cryptographic algorithms used on the IoT. In this study, we managed to block the linear cryptanalysis attack using a mathematically novel approach called Galois Field of the order (2$^{32}$), denoted by GF (2$^{32}$), and high irreducible polynomials were used to re-construct weak substitution boxes (S-Box) of mostly cryptographic algorithms used on IoT. It is a novel approach because no one has ever used GF (2$^{32}$) and highly irreducible polynomials to block linear cryptanalysis attacks on the most commonly used cryptographic algorithms. The most commonly used cryptographic algorithms on the IoT are Advanced Encryption Standard (AES), BLOWFISH, CAMELLIA, CAST, CLEFIA, Data Encryption Standard (DES), Modular Multiplication-based Block (MMB), RC5, SERPENT, and SKIPJACK. We assume that the reader of this paper has basic knowledge of the above algorithms.

**Keywords:** Internet of Things; cryptography; linear cryptanalysis attack; Galois Field; long irreducible polynomials; Advanced Encryption Standard (AES); BLOWFISH; CAMELLIA; CAST; CLEFIA; Data Encryption Standard (DES); Modular Multiplication-based Block (MMB); RC5; SERPENT; SKIPJACK

## 1. Introduction

In this study, the focus is mainly on the linear cryptanalysis attack of cryptographic algorithms commonly found and used in the Internet of Things (IoT). Linear cryptanalysis attacks are the biggest problem in the IoT and in cryptology itself. To block linear cryptanalysis attacks, we applied a mathematically novel approach called Galois Field (GF (2$^{32}$)) and high irreducible polynomials to be mapped on the S-Box (or on any building block attacked) of ten commonly used algorithms on the IoT. The most commonly used ten cryptographic algorithms are AES, BLOWFISH, CAMELLIA, CAST, CLEFIA, DES, MMB, RC5, SERPENT, and SKIPJACK. This approach is novel because no one has ever used GF (2$^{32}$) and polynomials to block linear cryptanalysis attacks on any cryptographic algorithms.

### 1.1. Motivation and Problem Statement of the Study

The main source of concern is intruders' use of linear cryptanalysis attacks on IoT cryptographic algorithms to discover the secret key [1–39]. An attacker employs linear

cryptanalysis attacks on IoT cryptographic algorithms by exploiting the low number of output bits of S-Boxes [1–39]. On the S-Boxes, an attacker can easily attack cryptographic algorithms with low-number output bits [9] (p. 21). On S-Boxes with less than 32 output bits, most well-known algorithms have a low number of output bits. AES and CAMELLIA, for example, have eight output bits of S-Boxes [10] (p. 16) and [11] (p. 18). S-Boxes are the four output bits of DES and SERPENT [12] (pp. 13–14) and [13] (p. 3). SKIPJACK has eight output bits, which are known as S-Boxes [14] (p. 8). Algorithms are vulnerable to attack due to their small size (low number of output bits) [9] (p. 21). If not addressed properly, the problem of a linear cryptanalysis attack can jeopardize the overall security of an IoT system. To combat these attacks, little has been achieved to increase the number of output bits on S-Boxes [1–39]. The goal of this research will be to solve the problem of linear cryptanalysis attacks. The GF ($2^{32}$) and long irreducible polynomials will be used to block linear cryptanalysis attacks on low-number output bits of S-Boxes of AES, BLOWFISH, CAMELLIA, CAST, CLEFIA, DES, MMB, RC5, SERPENT, and SKIPJACK.

### 1.2. Contribution of the Study

This study investigates how to make LAT more difficult for intruders to construct and more difficult to guess the key of cryptographic algorithms mapped with GF ($2^{32}$) and long irreducible polynomials. It has already been stated that the security of algorithms is dependent on the output bit size of S-Box; if the output bit size is small, attackers can easily attack the algorithm. That was the authors' previous hypothesis, but they did not know how plausible it was. We mapped GF ($2^{32}$) and long irreducible polynomials to increase the size of the S-Box output bits. GF ($2^{32}$) and long irreducible polynomials always produce 32-bit output when applied to S-Box or any algorithm building block.

### 1.3. Outline of the Study

The remaining sections are "Literature Review", "Linear Cryptanalysis Attack", "Theoretical Background of Our Novel Approach", "Research Methodology", "Results and Analysis", and "Conclusions and Future Work".

## 2. Literature Review

Algorithms used on IoT devices are attacked using linear cryptanalysis attacks. This attack reveals the cryptographic key to the intruder. Intruders use the key to obtain access to private data and information on the IoT system. Sakamura et al. [24] used the linear cryptanalysis attack on AES. Yongzhuang et al. [25] used the linear cryptanalysis attack on AES. The results of obtaining a key were successful on two rounds due to the way that the AES S-Box was constructed [25]. Blowfish is also one of the algorithms that suffers from linear cryptanalysis attacks. The small portions of keys were discovered after the linear differential cryptanalysis attack was applied to BLOWFISH because of the generated weak components (like P-arrays) [26]. P-array is the function defined on BLOWFISH; it has all the properties like S-Box [27]. The attack can only be detected if the weak key is used [26] (p. 20). CAMELLIA revealed its cryptographic key in round nine after being modified for the application of the linear cryptanalysis attack [27]. Keliher [28] attacked CAMELLIA using the linear cryptanalysis attack. Wang et al. [29] applied linear cryptanalysis to CAST; the cryptographic key was revealed on rounds 6 and 18 [29], and the experiment was conducted using the simplified S-Box. CLEFIA revealed its cryptographic key on 11, 12, 14, and 15, depending on the size of the inputs [31]. DES revealed its cryptographic key on rounds 3, 5, 8, 12, and 16 during the linear cryptanalysis attack [32]. MMB revealed its cryptographic key on all rounds during the linear cryptanalysis attack [33]. Kaliski and Yin [34] revealed the full cryptographic key of RC5 using the linear cryptanalysis attack [34]. SERPENT revealed its cryptographic key on round 11 during the linear cryptanalysis attack [35]. SKIPJACK reveals a certain bit of its cryptographic key from S-Box during the linear cryptanalysis attack [36] (p. 25). S-Boxes are the main building blocks of algorithms to give more chances to allow linear cryptanalysis attacks to be conducted [1–36,40–46].

The more S-Box is poorly constructed, the easier it is to conduct the linear cryptanalysis attack [1–36,40–46]. Little has been achieved to block linear cryptanalysis attacks on most cryptographic algorithms used on IoT devices [1–36,40–46]. In this study, we block linear cryptanalysis attacks by applying a mathematically novel approach of using the Galois Field of order **2³²**, denoted by GF (**2³²**), with the combination of generated irreducible polynomials to generate S-Boxes (or any function like S-Box) that will give 32-bit outputs. With the S-Box of 32-bit output, it is difficult to attack the algorithm using the linear cryptanalysis attack [29] (p. 429). A summary of the literature review is given in Table 1.

**Table 1.** Summary of literature review.

| Algorithm | Linear Cryptanalysis | Number of Rounds Attacked |
|---|---|---|
| AES | Yes [24,25] | 2 rounds [25] |
| BLOWFISH | Yes [26,27] | 18 rounds with week keys [26,27] |
| CAMELLIA | Yes [27,28] | 9 rounds [27] |
| CAST | Yes [29] | 6 and 18 rounds [29] |
| CLEFIA | Yes [31] | 11, 12, 14 and 15 rounds [31] |
| DES | Yes [32] | 3, 5, 8, 12 and 16 rounds [32] |
| MMB | Yes [33] | All rounds [33] |
| RC5 | Yes [34] | All rounds [34] |
| SERPENT | Yes [35] | 11 rounds [35] |
| SKIPJACK | Yes [36] | All rounds [36] |

### 2.1. Limitation of Study

Apart from the linear cryptanalysis attack, cryptography cannot guarantee information security. Additional techniques are required to protect against attacks, such as denial of service or complete system failure [24–26].

### 2.2. Definition of Internet of Things (IoT)

IoT is an upcoming network platform and a new paradigm of communication innovations of the future that keeps on connecting huge amounts of different devices in order to provide new proper services [40–45]. Devices are technological machines or things used to establish any communication when IoT is used, like smart cards, sensors, temperature monitors, radio frequency identification (RFID), and many more [40–45].

### 2.3. Definition of Cryptography

Cryptography is the mathematical procedure that changes plaintext (a readable message) into ciphertext (an unreadable scrambled message) and vice versa. The mathematical procedure of changing plaintext to ciphertext is called encryption, and changing ciphertext back to plaintext is called decryption [44,46]. In simple terms, cryptography is a desirable procedure to secure communications and data meant to block intruders or attackers from obtaining access to confidential information [44–46]. This mathematical procedure of encryption and decryption is called the cryptographic algorithm. For example, AES, BLOWFISH, CAMELLIA, CAST, CLEFIA, DES, MMB, RC5, SERPENT, and SKIPJACK.

### 2.4. Linear Cryptanalysis Attack (also Known as Known Plaintext Attack)

Linear cryptanalysis (or known plaintext attack) is a strong technique introduced by Matsui in 1993 to attack cryptographic algorithms [1]. The attack was first implemented on the cryptographic algorithm called the Data Encryption Standard (DES), but an untimely adaptation of the linear cryptanalysis attack, introduced by Matsui and Yamagishi, was previously used to successfully attack FEAL in 1992 [2]. The attacker analyzes the linear probability relations (linear approximations) of known plaintext to search for a secret key.

Cryptographic algorithms routinely use non-linear S-Boxes in their structures [1–3]. In DES, the only non-linear building blocks are the S-boxes [3]. The rest of the building blocks are linear and can be easily attacked [3]. A linear probability relation (linear approximation) of known plaintext is used to obtain the secret key and is calculated by XORing many pairs of plaintext, and the results are tabled for analysis [4]. Attackers do not attack the entire algorithm at once [4]. In the linear cryptanalysis attack, the attacker first analyzes the linear vulnerabilities of an S-Box. For example, let us consider the simplified S-Box of DES shown in Table 1, where the input bits are X = [$X_1$ $X_2$ $X_3$ $X_4$] and the output bits are Y = [$Y_1$ $Y_2$ $Y_3$ $Y_4$]. The attacker examined all possibilities of the events that S-Box conducted. Every linear approximation can be calculated to discover its functionality by examining the probability for each event using Equation (1) [5].

$$Probability[\ X_1 \oplus\ X_2 \oplus \ldots \oplus X_U \oplus Y_1 \oplus Y_2 \oplus \ldots \oplus Y_V = 0] = P \tag{1}$$

From Equation (1), if is much greater than half (1/2), there is a high probability of occurrence to guess the key, whereas if is much lesser than half, the S-Box has a low probability of occurrence [1]. Therefore, the probability bias of the S-Box can be calculated using Equation (1) [2–4]. Figure 1 shows the four input bits X = [$X_1$ $X_2$ $X_3$ $X_4$] and the four output bits Y = [$Y_1$ $Y_2$ $Y_3$ $Y_4$] of the simplified S-Box of DES defined in Table 2. The input and output bits are used to define the S-Box size. The S-Box size in Figure 1 and Table 2 is a 4 × 4 S-Box.

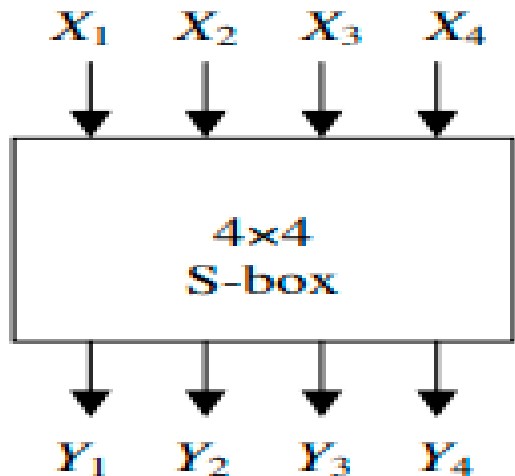

**Figure 1.** Sample of S-Box [4].

**Table 2.** Simplified S-Box of DES [5].

| X | 0 | 1 | 2 | 3 | 4 | 5 | 6 | 7 | 8 | 9 | A | B | C | D | E | F |
|---|---|---|---|---|---|---|---|---|---|---|---|---|---|---|---|---|
| S(X) = Y | E | 4 | D | 1 | 2 | F | B | 8 | 3 | A | 6 | C | 5 | 9 | 0 | 7 |

For example, From Table 2 and Figure 1, S-Box is 4 × 4, meaning that it has four bits of input and four bits of output. To calculate the linear probability, if an event of $X_1 \oplus X_4 \oplus Y_2 = 0$ is analyzed, Table 3 can be drawn, and a total of eight zeros are counted. See Table 3. Therefore, from Equation (2), derived from Equation (1),

$$Probability[\ X_1 \oplus\ X_4 \oplus Y_2 = 0] = \frac{8}{16} = \frac{1}{2} = PL \tag{2}$$

**Table 3.** Calculating linearity when $X_1 \oplus X_4 \oplus Y_2 = 0$ [6].

| $X_1$ | $X_2$ | $X_3$ | $X_4$ | $Y_1$ | $Y_2$ | $Y_3$ | $Y_4$ | |
|-------|-------|-------|-------|-------|-------|-------|-------|---|
| 0 | 0 | 0 | 0 | 1 | 1 | 1 | 0 | 1 |
| 0 | 0 | 0 | 1 | 0 | 1 | 0 | 0 | 0 |
| 0 | 0 | 1 | 0 | 1 | 1 | 0 | 1 | 1 |
| 0 | 0 | 1 | 1 | 0 | 0 | 0 | 1 | 1 |
| 0 | 1 | 0 | 0 | 0 | 0 | 1 | 0 | 0 |
| 0 | 1 | 0 | 1 | 1 | 1 | 1 | 1 | 0 |
| 0 | 1 | 1 | 0 | 1 | 0 | 1 | 1 | 0 |
| 0 | 1 | 1 | 1 | 1 | 0 | 0 | 0 | 1 |
| 1 | 0 | 0 | 0 | 0 | 0 | 1 | 1 | 1 |
| 1 | 0 | 0 | 1 | 1 | 0 | 1 | 0 | 0 |
| 1 | 0 | 1 | 0 | 0 | 1 | 1 | 0 | 0 |
| 1 | 0 | 1 | 1 | 1 | 1 | 0 | 0 | 1 |
| 1 | 1 | 0 | 0 | 0 | 1 | 0 | 1 | 0 |
| 1 | 1 | 0 | 1 | 1 | 0 | 0 | 1 | 0 |
| 1 | 1 | 1 | 0 | 0 | 0 | 0 | 0 | 1 |
| 1 | 1 | 1 | 1 | 0 | 1 | 1 | 1 | 1 |

The denominator value of 16 is calculated from the size of the S-Box, which is $4 \times 4 = 16$. Alternatively, the denominator value can also be calculated using ($2^4 = 16$), where 4 is the number of output bits [1–6]. Therefore,

$$Probability\ bias = PL - \frac{1}{2} = 0 \tag{3}$$

Let us take another example: when the event of $X_3 \oplus X_4 \oplus Y_1 \oplus Y_4 = 0$ is analyzed, the truth table of the simplified S-Box of DES is defined in the table with input bits $X = [X_1\ X_2\ X_3\ X_4]$ and the output bits $Y = [Y_1\ Y_2\ Y_3\ Y_4]$ drawn in Table 4. The results where $X_3 \oplus X_4 \oplus Y_1 \oplus Y_4$ are listed in the last column in red. When $X_3 \oplus X_4 \oplus Y_1 \oplus Y_4 = 0$, only two occurrences are found. Refer to Table 4. Therefore, using Equation (4) derived from Equation (1),

$$Probability[\ X_3\ \oplus\ X_4 \oplus Y_1 \oplus Y_4 = 0] = \frac{2}{16} = \frac{1}{8} \tag{4}$$

Therefore,

$$Probability\ bias = PL - \frac{3}{8} = -0.375 \tag{5}$$

Though sometimes the minus value is found, the attacker is always interested in the absolute value of the probability bias [4]. After the entire possible event is analyzed, the attacker then constructs a table called the Linear Approximation Table (LAT) [1–6], using Equations (1) and (2) and their truth table. Figure 2 shows all possible linear approximations of the simplified S-Box of DES given in Table 2 and Figure 1. After the LAT is constructed, the attacker uses a simple mathematical and statistical guess in a very short space of time to discover the key of the cryptographic algorithm [4,5]. No LAT, no linear approximation; therefore, no linear cryptanalysis attack [7,8]. With the use of the LAT, an attacker traces and attacks the entire algorithm by tracing the changes in the bits of the entire algorithm, like what is outlined in Figure 3. In Figure 3, the red lines are indications of the trace probability of the secret key using LAT.

**Table 4.** Calculating linearity when $X_3 \oplus X_4 \oplus Y_1 \oplus Y_4 = 0$ [6].

| $X_1$ | $X_2$ | $X_3$ | $X_4$ | $Y_1$ | $Y_2$ | $Y_3$ | $Y_4$ | |
|-------|-------|-------|-------|-------|-------|-------|-------|---|
| 0 | 0 | 0 | 0 | 1 | 1 | 1 | 0 | 1 |
| 0 | 0 | 0 | 1 | 0 | 1 | 0 | 0 | 1 |
| 0 | 0 | 1 | 0 | 1 | 1 | 0 | 1 | 0 |
| 0 | 0 | 1 | 1 | 0 | 0 | 0 | 1 | 1 |
| 0 | 1 | 0 | 0 | 0 | 0 | 1 | 0 | 0 |
| 0 | 1 | 0 | 1 | 1 | 1 | 1 | 1 | 1 |
| 0 | 1 | 1 | 0 | 1 | 0 | 1 | 1 | 1 |
| 0 | 1 | 1 | 1 | 1 | 0 | 0 | 0 | 1 |
| 1 | 0 | 0 | 0 | 0 | 0 | 1 | 1 | 1 |
| 1 | 0 | 0 | 1 | 1 | 0 | 1 | 0 | 1 |
| 1 | 0 | 1 | 0 | 0 | 1 | 1 | 0 | 1 |
| 1 | 0 | 1 | 1 | 1 | 1 | 0 | 0 | 1 |
| 1 | 1 | 0 | 0 | 0 | 1 | 0 | 1 | 1 |
| 1 | 1 | 0 | 1 | 1 | 0 | 0 | 1 | 1 |
| 1 | 1 | 1 | 0 | 0 | 0 | 0 | 0 | 1 |
| 1 | 1 | 1 | 1 | 0 | 1 | 1 | 1 | 1 |

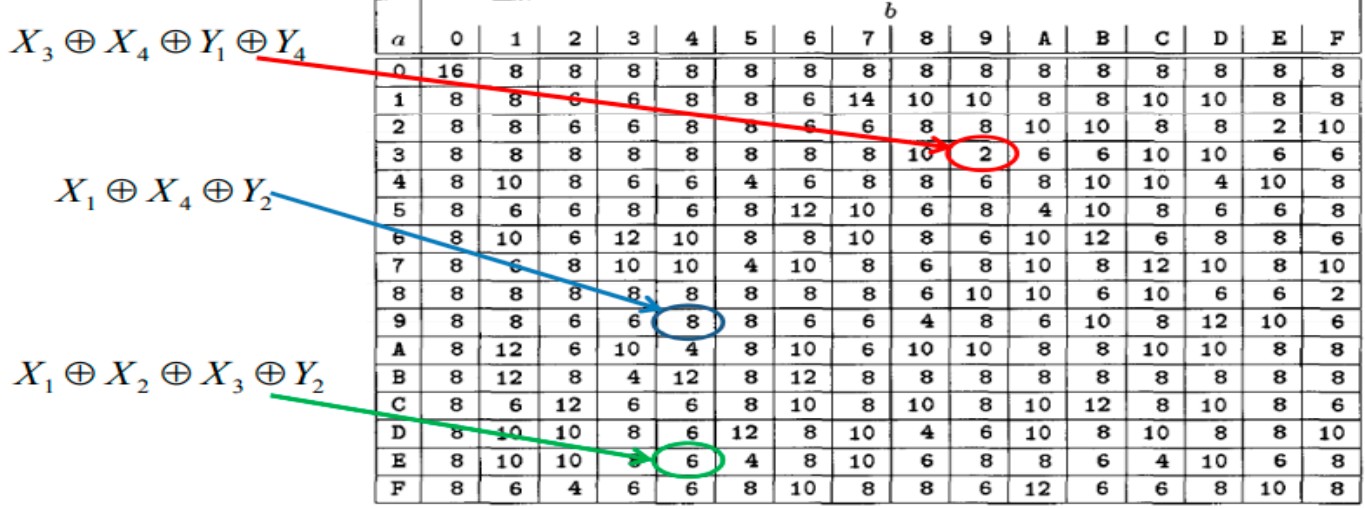

**Figure 2.** Linear Approximation Table (LAT) of simplified S-Box of DES when zeros are counted [6].

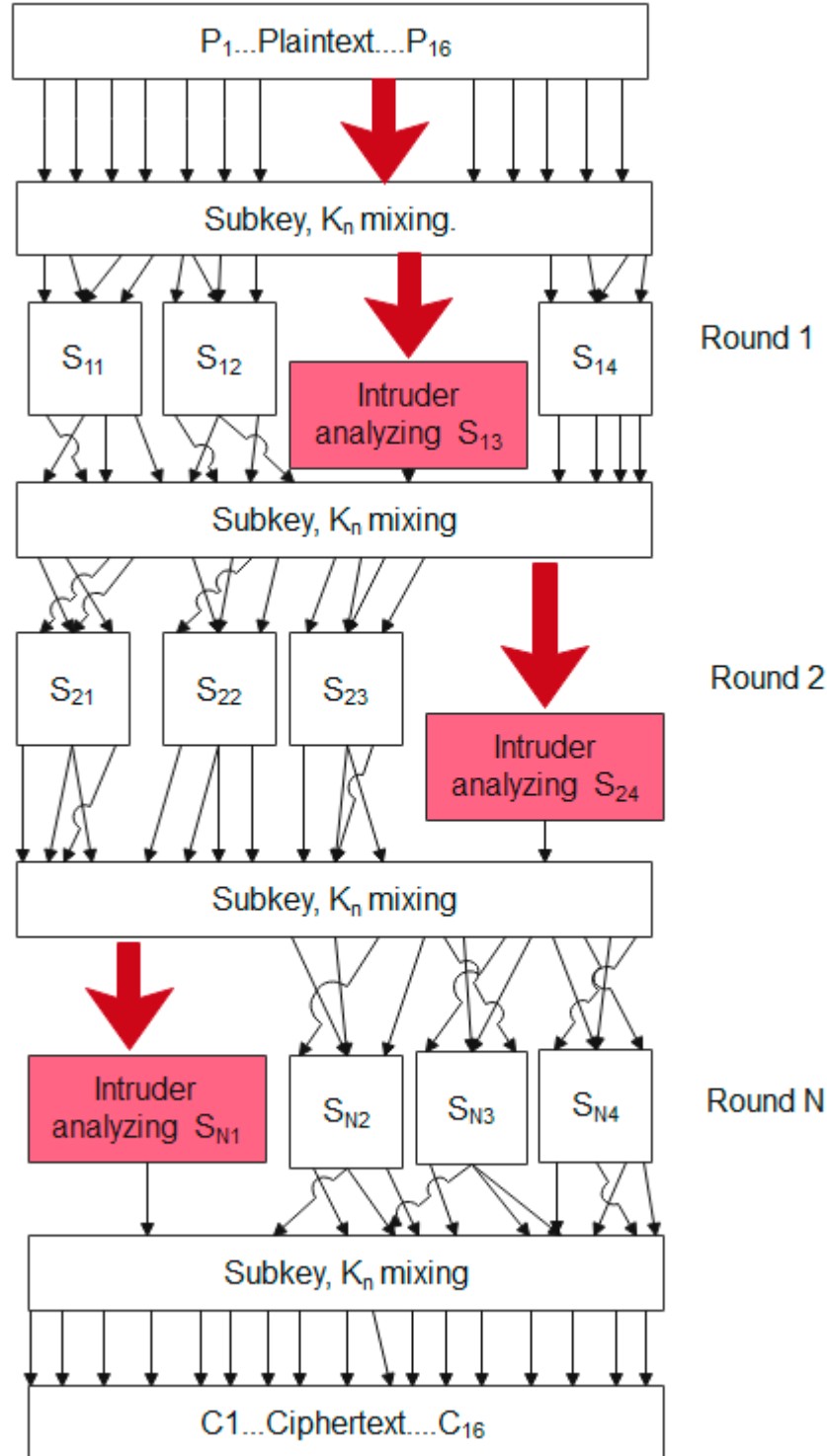

**Figure 3.** Tracing the entire cryptographic algorithm to obtain a key [5].

### 3. Theoretical Background of Our Novel Approach

In this study, the proposed procedure (novel approach), which is believed to cater to less probability, give a big table of LAT, and bring confusion to the attacker, is the Galois Field of highest order, which will yield 32 output bits compared to the minimum output bits of known S-Boxes. Hosseinkhani et al. [9] (p. 21) indicated that the S-Boxes with more output bits provide high security. In cryptography, the size of the S-Box is defined using input bits (let us say the number of input bits is N) and output bits (let us say the number

of output bits is M); therefore, the size of the S-Box is NxM. In this proposal, the highest chosen order of Galois Field is $2^{32}$. A Galois Field of order $2^{32}$ is mathematically represented by **GF ($2^{32}$)**; this means that the size of an S-Box constructed by **GF ($2^{32}$)** will be Nx32 (that is, N input bits and 32 output bits). Most algorithms used on IoT (like AES, BLOWFISH, CAMELLIA, CAST, CLEFIA, DES, MMB, RC5, SERPENT, and SKIPJACK) have S-Boxes that have fewer output bits than 32, making them vulnerable to attackers [9]. Hosseinkhani et al. [9] (p. 21) indicated that the S-Boxes with fewer output bits provide less security. The size of tables like LAT is determined by the size of the S-Box. For example, if the simplified DES S-Box size is $4 \times 4$, then the LAT size will have 16 columns $\times$ 16 rows, which is calculated as $2^4 \times 2^4$. Refer to Table 4. Therefore, an S-Box of $32 \times 32$, for example, will give a LAT of the size of $2^{32} \times 2^{32}$, which is 4,294,967,296 (number of columns) $\times$ 4,294,967,296 (number of rows). With this increase in events, the attacker is believed to have low chances or probability of guessing a key [15]. This is also supported by statistical theory (the probability of guessing a head from a coin is 1/2 and the probability of guessing a space ace from a deck card is 1/(56)) [16,17]. With the size ($2^{32}$ = 4,294,967,296) probability table, attackers will need extra time (or struggle) to construct and analyze tables such as LAT, unlike before when the size of an S-Box was $4 \times 4$. After the construction of an S-Box of Nx32 using **GF ($2^{32}$)** and a high irreducible polynomial, where N and 32 are the number of input and output bits, respectively, the developer can even construct an S-Box of $4 \times 32$, $6 \times 32$, $8 \times 32$, and $16 \times 32$ by extracting entities from the original Nx32 S-Box [9–17]. If N = 4, 6, 8, or 16, a new S-Box of $4 \times 32$, $6 \times 32$, $8 \times 32$, and $16 \times 32$ constructed by GF ($2^{32}$), respectively, would give complicated probability tables of LAT of $2^4 \times 2^{32}$, $2^6 \times 2^{32}$, $2^8 \times 2^{32}$, and $2^{16} \times 2^{32}$, respectively. The number of rows still needed to construct LAT will be $2^{32}$ = 4,294,967,296. The other aspect is driven from a statistical point of view: if the entities of the event are greater, it is more difficult to guess an entity to be true. Therefore, using a higher-order Galois Field, the probability of guessing the key will be too low, unlike before when the size of an S-Box was $4 \times 4$. Little has been achieved to reconstruct, replace, and strengthen the S-Boxes of ten cryptographic algorithms, which are commonly used on the IoT. In this proposal, reconstruction of algorithms is going to be performed, and reconstruction of small output S-Boxes to 32-bit output is going to be developed using **GF ($2^{32}$)**. The proposed procedure to construct a strong algorithm is based on Galois Field theory.

*Theory of Galois Field (GF ($p^q$))*

Evariste Galois, who was a great French algebraist who died at the age of 20 years, discovered Galois Field theory [18–23]. According to Galois Field theory, if p is a prime number, then it is also possible to define a field with $p^q$ elements for any q. A field is a set function F with two composition laws, plus (+) and multiplication (*), such that

a.  (F, +) is a commutative [18–23]. This means that any element of a set F (for example, the *a*, *b*, *and c* elements of F), when added, holds the following property [18]:

$$a + b + c = c + a + b = b + c + a \tag{6}$$

b.  ($F^X$, *) is a commutative [19]. This means that any element of a set F (for example, the *a*, *b*, *and c* elements of F), when multiplied, holds the following property [18–23]:

$$a * b * c = c * a * b = b * c * a \tag{7}$$

c.  The distributive law holds, and each element has an inverse [18]. This means that any element of a set F (for example, the *a*, *b*, *and c* elements F), when computed, holds the following property [18–23]:

$$a * (b + c) = a * b + a * c \tag{8}$$

The total number of elements in the field is defined as the order of a field [20]. The order of GF ($p^q$) is $p^q$. For each Galois Field (GF ($p^q$)), there exists a prime number (irreducible polynomial) of base p with element q. An irreducible polynomial is a polynomial that cannot be factorized or divided into more than one segment except by 1 or itself [9–23]. For example, GF ($2^2$), has an irreducible polynomial of $x^1 + 1 = 11_{binary} = 3_{decimal}$ [21] (p. 385). This means that from $2^0$ to $2^2$, the only number that cannot be factored or divided except by 1 or itself is 3. For GF ($2^3$), there exist two irreducible polynomials: $x^3 + x + 1 = 1011_{binary} = 11_{decimal}$ and $x^3 + x^2 + 1 = 1101_{binary} = 13_{decimal}$ [9] (p. 385). Therefore, in GF ($\mathbf{2^{32}}$), there exist at least six long irreducible polynomials as follows [23] (pp. 25–26), [18]:

$$x^{32} + x^{22} + x^2 + x + 1 = 80200007_{hexidecimal} = 2149580807_{decimal} \qquad (9)$$

$$x^{32} + x^{22} + x^{21} + x^{20} + x^{18} + x^{17} + x^{15} + x^{13} + x^{12} + x^{10} + x^8 + x^6 + x^4 + x^1 \\ +1 = 10076B553_{hexidecimal} \qquad (10)$$

$$x^{32} + x^{23} + x^{17} + x^{16} + x^{14} + x^{10} + x^8 + x^7 + x^6 + x^5 + x^3 + 1 = f_1(x) \qquad (11)$$

$$x^{32} + x^{26} + x^{23} + x^{22} + x^{16} + x^{12} + x^{11} + x^{10} + x^8 + x^7 + x^5 + x^4 + x^2 + x^1 + 1 \\ = f_2(x) \qquad (12)$$

$$x^{32} + x^{27} + x^{26} + x^{25} + x^{24} + x^{23} + x^{22} + x^{17} + x^{13} + x^{11} + x^{10} + x^9 + x^8 + x^7 + \\ x^2 + x^1 + 1 = 10FC22F87_{hexadecimal} = 4559351687_{decimal} \qquad (13)$$

$$x^{32} + x^{28} + x^{19} + x^{18} + x^{16} + x^{14} + x^{11} + x^{10} + x^9 + x^6 + x^5 + x^1 + 1 = f_3(x) \qquad (14)$$

It is already stated that GF ($\mathbf{2^{32}}$) is the chosen mathematical procedure to be used to generate S-Boxes with an output of 32 bits. It is noted that a polynomial $x^{32} + x^{22} + x^{21} + x^{20} + x^{18} + x^{17} + x^{15} + x^{13} + x^{12} + x^{10} + x^8 + x^6 + x^4 + x^1 + 1 = 10076B553_{hexidecimal}$ and a polynomial $x^{32} + x^{27} + x^{26} + x^{25} + x^{24} + x^{23} + x^{22} + x^{17} + x^{13} + x^{11} + x^{10} + x^9 + x^8 + x^7 + x^2 + x^1 + 1 = 10FC22F87_{hexadecimal} = 4559351687_{decimal}$ and are long irreducible polynomials. To recall the simplified S-Box of DES from Table 2, see Table 5. We multiplied each of the four output bits of the simplified S-Box of DES by $10076B553_{hexidecimal}$, then added the product to $10FC22F87_{hexadecimal}$; the sum will be multiplied by modulus ($\mathbf{2^{32}}$). Modulus ($\mathbf{2^{32}}$) was used to quantify (make sure) that the entity generated has a 32-bit output. For example, let us convert the simplified S-Box of DES, which is $4 \times 4$ (four inputs and four outputs), to $4 \times 32$ using GF ($\mathbf{2^{32}}$) and long irreducible polynomials. Refer to Equations (15) and (16).

**Table 5.** Recall of Table 2 of simplified S-Box of DES [5].

| X | 0 | 1 | 2 | 3 | 4 | 5 | 6 | 7 | 8 | 9 | A | B | C | D | E | F |
|---|---|---|---|---|---|---|---|---|---|---|---|---|---|---|---|---|
| S(X) = Y | E | 4 | D | 1 | 2 | F | B | 8 | 3 | A | 6 | C | 5 | 9 | 0 | 7 |

The first entity is $E_{hexadecimal} = 14_{decimal}$ and is 4 outputted bits; to convert it to 32 bits, $E_{hexadecimal}$ is multiplied by $10076B553_{hexidecimal}$, the product will be added with $10FC22F87_{hexadecimal}$ and then the sum will be modulated by mod ($\mathbf{2^{32}}$) = Modulus ($100000000_{hexadecimal}$). That is

$$S(0) \\ = (E_{hexadecimal} * 10076B553_{hexadecimal} \\ + 10FC22F87_{hexadecimal}) \; mod \; (100000000)_{hexadecimal} \\ = 16401A11_{hexadeciaml} \qquad (15)$$

$$S(1)$$
$$= (4 * 10076B553_{hexadecimal}$$
$$+ 10FC22F87_{hexadecimal}) \; mod \; (100000000)_{hexadecimal}$$
$$= 119D04D3_{hexadeciaml}$$

(16)

Continuing to convert all 4 output bits to 32 output bits of the simplified S-Box, the new S-Box will be represented by Table 6.

**Table 6.** New S-Box generated by GF ($2^{32}$) and long irreducible polynomial of simplified DES's S-Box.

| X | S(X) = Y |
|---|----------|
| 0 | 16401A11 |
| 1 | 119D04D3 |
| 2 | 15C964DE |
| 3 | 1038E4DA |
| 4 | 10AF9A2D |
| 5 | 16B6CF6A |
| 6 | 14DBFA18 |
| 7 | 1377DAIF |
| 8 | 11264780 |
| 9 | 146544C5 |
| A | 12846F79 |
| B | 1552AF6B |
| C | 1213BA26 |
| D | 13EE8F72 |
| E | 0FC22787 |
| F | 130124CC |

All S-Boxes of ten algorithms commonly used on IoT (AES, BLOWFISH, CAMELLIA, CAST, CLEFIA, DES, MMB, RC5, SERPENT, and SKIPJACK) will be reconstructed using the GF ($2^{32}$) method to strengthen them against linear cryptanalysis attacks. Algorithms will also be reconstructed to suit newly generated S-Boxes.

## 4. Research Methodology

The research methodology was based on preventing linear cryptanalysis attacks by mapping **GF** ($2^{32}$) and long irreducible polynomials on the building blocks that we found to be the weakest links during a linear cryptanalysis attack. The research was conducted as follows:

i.     We collected all ten algorithms (AES, BLOWFISH, CAMELLIA, CAST, CLEFIA, DES, MMB, RC5, SERPENT, and SKIPJACK) from different IoT devices using FileDisassembler and analyzed all the .dll cryptographic files using dotPeek and iLSpy.

ii.    We checked all ten algorithms' correctness using test vectors from their developers. Test vectors are sets of inputs and outputs for any system to check the system's correctness [26]. For example, test vectors of the AES algorithm are shown in [26] (p. 35). For more information about test vectors, refer to Appendix A.

iii.   We analyzed the way all algorithms are attacked by intruders using the linear cryptanalysis attack method.

iv.    We mapped or applied **GF ($2^{32}$)** and long irreducible polynomials to the building blocks that are the weakest link during the linear cryptanalysis attack.

v.      We checked whether it was still possible for the linear cryptanalysis attack to be successful after **GF ($2^{32}$)** and long irreducible polynomials were applied. If it was still possible, we went back to steps (iv) and (v).

vi.      If the linear cryptanalysis attack is blocked on steps (iii), (iv), and (v), then we accept and rebuild a new algorithm mapped with **GF ($2^{32}$)** and long irreducible polynomials as an algorithm that is resistant to the linear cryptanalysis attack.

For example, we investigated how to make LAT more difficult for attackers to construct and more cumbersome to guess the key of cryptographic algorithms mapped with **GF ($2^{32}$)** and long irreducible polynomials. It is already stated that the security of algorithms relies on the output bit size of S-Box; if the output bit size is small, it is easy for attackers to attack algorithms. That is the theory we had before, but we did not know how possible it was. We mapped **GF ($2^{32}$)** and long irreducible polynomials to increase the size of the output bits of S-Boxes. **GF ($2^{32}$)** and long irreducible polynomials always yield 32 bits of output when applied to S-Box or any building block of an algorithm. The research methodology is summarized by the schematic diagram in Figure 4.

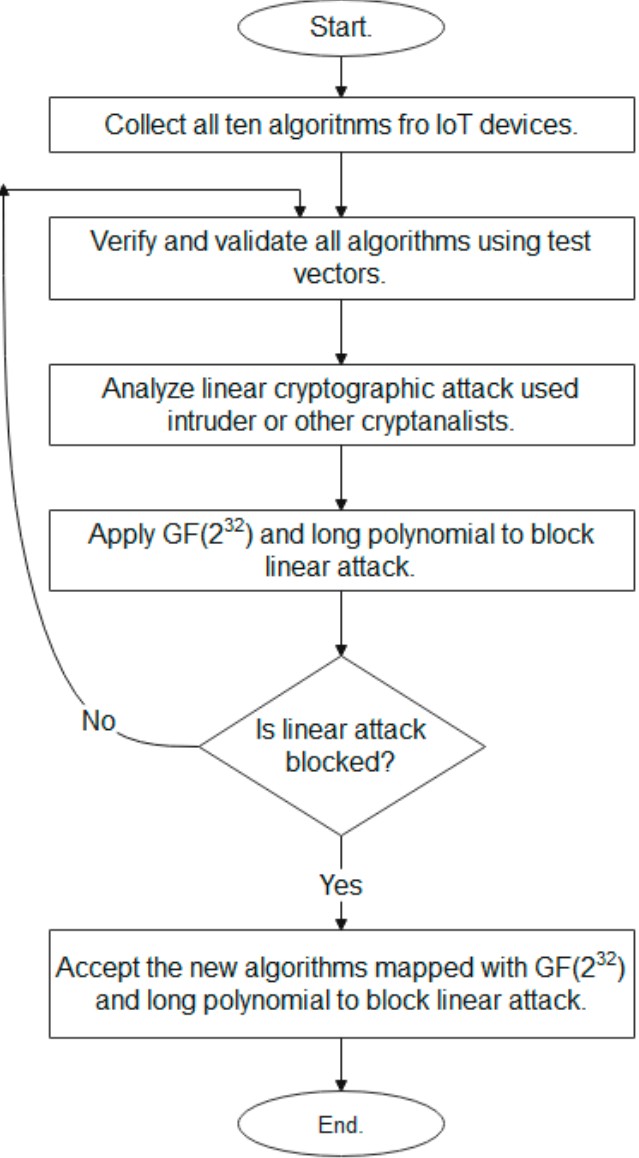

**Figure 4.** Schematic diagram of research methodology.

## 5. Results and Analysis

On AES, we found that linear cryptanalysis attacks are possible on rounds one and two only. Attacks are impossible on rounds three and above. The main building block that makes rounds one and two possible for linear cryptanalysis attacks is the S-Box. The S-Box of AES is $8 \times 8$, meaning that it has 8 inputs and 8 outputs. We found that it is easy to construct a LAT table using the AES S-Box. The LAT of the AES S-Box is a table of $2^8 \times 2^8$ matrixes with probability entities for guessing a key. We wrote a C++ program to construct the LAT of the AES S-Box. We confirmed that it is possible to attack the AES algorithm using LAT after analyzing the procedure. To block linear attack, we mapped GF ($2^{32}$) and its long irreducible polynomial to obtain a new S-Box with 32-bit output. The new S-Box for AES was constructed. We tried to construct the LAT of a new S-Box, which is supposed to be a $2^8 \times 2^{32}$ matrix. That is a $256 \times 4{,}294{,}967{,}296$ matrix. We found that it is infeasible to construct the LAT of a new AES S-Box with an output of 32 bits because of the maximum size allocation. We tried to use array $2^{32} = 4{,}294{,}967{,}296$ size, and we found that we also have to include input $2^8 = 256$. The program for constructing LAT on a new S-Box crashed before LAT was constructed due to a lack of memory. The computation of $2^{32}$ needs more than $2^{64}$ memory allocations, which is impossible. Iwata et al. [37] (p. 121) also confirmed that it is infeasible to construct a probability table (like LAT) for a 32-bit output S-Box because it needs a memory of $2^{64}$, which is impossible. To achieve probabilities of guessing a key from a 32-bit output, S-Box is impossible [38] (p. 9). 32-bit S-Boxes are robust to linear cryptanalysis attacks due to the cost of computing LAT due to the memory required [39] (p. 179). Therefore, we managed to block and prevent LAT construction using GF ($2^{32}$) and its long irreducible polynomial. GF ($2^{32}$) and long irreducible polynomials always yield 32-bit output when applied to S-Box or any building block of an algorithm. No LAT, no linear cryptanalysis attack. Therefore, we managed to secure AES against the linear cryptanalysis attack. For Blowfish, the linear cryptanalysis attack was performed on P-arrays rather than S-boxes. S-Boxes of Blowfish already have 32-bit output, so attackers avoided S-Boxes and attacked P-array because P-array is weakly generated [26]. P-array is generated from PI ratio = 22/7; all digits are taken from the string of P1 [26] (p. 21). We applied GF ($2^{32}$) and the long irreducible polynomial to the P-array, and we tried to do the linear cryptanalysis attack using the same steps used on [26]. We found it to be impossible because we destroyed the weakness of the P-array and the nature of the PI derivation, which was the advantage of the attacker in running the linear cryptanalysis attack easily. We found that CAST already has 32-bit out S-Boxes; attackers reduced output CAST S-Boxes [45]. We applied GF ($2^{32}$) and a long irreducible polynomial to the original 32-bit output S-Boxes and tried to apply a linear cryptanalysis attack. We found that it was impossible to guess the key. We found that RC5 and MMB have no S-Boxes. We used the new AES S-Box on RC5 and MMB to block linear cryptanalysis attacks since the new AES S-Box has already been developed and proven to be strong against linear cryptanalysis attacks. For the rest of the algorithms, like DES, CAMELLIA, SERPENT, etc., we applied GF ($2^{32}$) and its irreducible polynomial. We found that all are resistant to linear cryptanalysis attacks due to the memory required to construct LAT. No LAT, no linear cryptanalysis attack. We rewrote all ten algorithms with new components mapped to GF ($2^{32}$) and long irreducible polynomials. We tested whether all ten algorithms were encrypting and decrypting with new 32-bit output components. All were working according to our expectations.

To prove that we conducted all procedures of linear cryptanalysis attack used to attack algorithms using the LAT, we give an executable file of the simplified S-Box defined in Figure 2. We analyzed how Figure 2 is theoretically constructed and wrote our experimental C++ code for verification. Refer to Figures 2 and 5. Figures 2 and 5 have the same entities. Figure 2 is our theoretical LAT, and Figure 5 is our experimental LAT executed by our C++ LAT code.

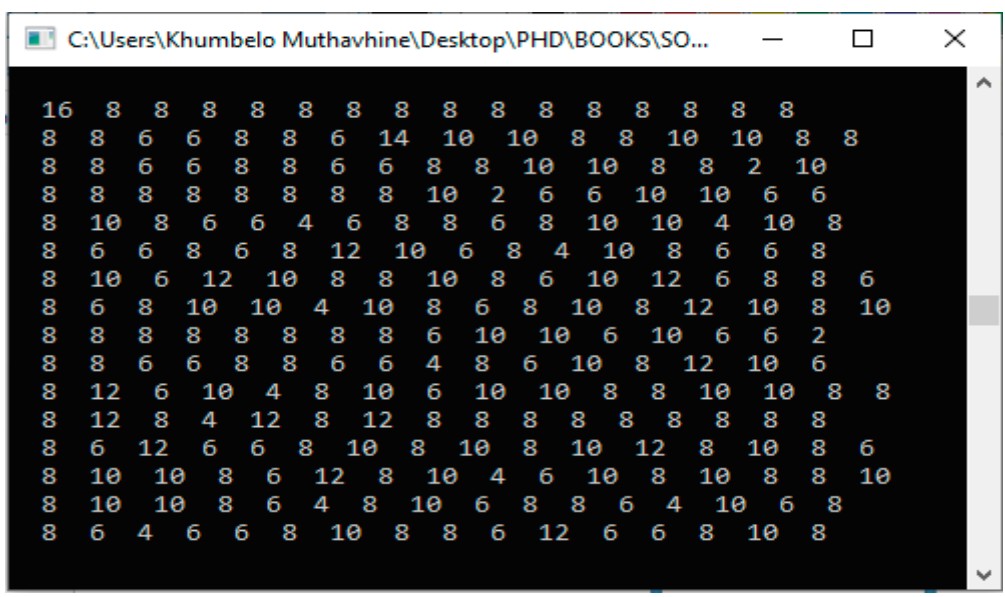

**Figure 5.** C++ output of LAT's construction.

Table 7 compares theoretical and experimental results before and after GF ($2^{32}$) was applied. Table 8 shows the results of time complexity before and after the GF ($2^{32}$) was applied. Table 9 shows the results of LAT constructability before and after the GF ($2^{32}$) was applied.

**Table 7.** Results of linear attack before and after the GF ($2^{32}$) was applied.

| Algorithm | Linear Cryptanalysis before GF ($2^{32}$) Was Applied | Number of Rounds Attacked before GF ($2^{32}$) Was Applied | Linear Cryptanalysis after GF ($2^{32}$) Was Applied | Number of Rounds Attacked after GF ($2^{32}$) Was Applied |
| --- | --- | --- | --- | --- |
| AES | Yes [24,25] | 2 rounds [25] | No | None |
| BLOWFISH | Yes [26,27] | 16 rounds with week keys [26,27] | No | None |
| CAMELLIA | Yes [27,28] | 9 rounds [27] | No | None |
| CAST | Yes [29] | 6 and 18 rounds [29] | No | None |
| CLEFIA | Yes [31] | 11, 12, 14 and 15 rounds [31] | No | None |
| DES | Yes [32] | 3, 5, 8, 12 and 16 rounds [32] | No | None |
| MMB | Yes [33] | All rounds [33] | No | None |
| RC5 | Yes [34] | All rounds [34] | No | None |
| SERPENT | Yes [35] | 11 rounds [35] | No | None |
| SKIPJACK | Yes [36] | All rounds [36] | No | None |

**Table 8.** Results of time complexity before and after the GF ($2^{32}$) was applied.

| Algorithm | Attack Time Complexity before GF ($2^{32}$) Was Applied | Number of Rounds Attacked before GF ($2^{32}$) Was Applied | Attack Time Complexity after GF ($2^{32}$) Was Applied | Computation of LAT after GF ($2^{32}$) Was Applied |
| --- | --- | --- | --- | --- |
| AES | $2^{176}$ [47] | 2 rounds [25] | C++ program crashed before completing execution | Impossible, no ordinary computer can compute and store a $2^{32}$ LAT simultaneously [37] (p. 121) |

**Table 8.** *Cont.*

| Algorithm | Attack Time Complexity before GF ($2^{32}$) Was Applied | Number of Rounds Attacked before GF ($2^{32}$) Was Applied | Attack Time Complexity after GF ($2^{32}$) Was Applied | Computation of LAT after GF ($2^{32}$) Was Applied |
|---|---|---|---|---|
| BLOWFISH | $2^{14}$ [26] | 16 rounds with week keys [26,27] | C++ program crashed before completing execution | Impossible, no ordinary computer can compute and store a $2^{32}$ LAT simultaneously [37] (p. 121). |
| CAMELLIA | $2^{110}$ [48] | 9 rounds [27] | C++ program crashed before completing execution | Impossible, no ordinary computer can compute and store a $2^{32}$ LAT simultaneously [37] (p. 121). |
| CAST | $2^{-12.91}$ [29] | 6 and 18 rounds [29] | C++ program crashed before completing execution | Impossible, no ordinary computer can compute and store a $2^{32}$ LAT simultaneously [37] (p. 121). |
| CLEFIA | $2^{244.2}$ [30] | 11, 12, 14 and 15 rounds [31] | C++ program crashed before completing execution | Impossible, no ordinary computer can compute and store a $2^{32}$ LAT simultaneously [37] (p. 121). |
| DES | $2^{43}$ [49] | 3, 5, 8, 12 and 16 rounds [32] | C++ program crashed before completing execution | Impossible, no ordinary computer can compute and store a $2^{32}$ LAT simultaneously [37] (p. 121). |
| MMB | $2^{40}$ [50] | All rounds [33] | C++ program crashed before completing execution | Impossible, no ordinary computer can compute and store a $2^{32}$ LAT simultaneously [37] (p. 121). |
| RC5 | $2^{36}$ [51] | All rounds [34] | Infinity. C++ program crashed. | Impossible, no ordinary computer can compute and store a $2^{32}$ LAT simultaneously [37] (p. 121). |
| SERPENT | $2^{80}$ [52] | 11 rounds [35] | C++ program crashed before completing execution | Impossible, no ordinary computer can compute and store a $2^{32}$ LAT simultaneously [37] (p. 121). |
| SKIPJACK | $2^{49}$ [53] | All rounds [36] | C++ program crashed before completing execution | Impossible, no ordinary computer can compute and store a $2^{32}$ LAT simultaneously [37] (p. 121). |

**Table 9.** Results of LAT constructability before and after the GF ($2^{32}$) was applied.

| Algorithm | Possibility of Linear Attack before GF ($2^{32}$) Was Applied | Remarks | Possibility of Linear Attack after GF ($2^{32}$) Was Applied |
|---|---|---|---|
| AES | Possible because LAT was constructible [47] | Linear cryptanalysis attack relies on LAT. | Impossible because LAT was not constructible, due to memory constraints [39] (p. 176). |
| BLOWFISH | Possible because LAT was constructible [26] | Linear cryptanalysis attack relies on LAT. | Impossible because LAT was not constructible, due to memory constraints [39] (p. 176). |
| CAMELLIA | Possible because LAT was constructible [48] | Linear cryptanalysis attack relies on LAT. | Impossible because LAT was not constructible, due to memory constraints [39] (p. 176). |
| CAST | Possible because LAT was constructible [29] | Linear cryptanalysis attack relies on LAT. | Impossible because LAT was not constructible, due to memory constraints [39] (p. 176). |
| CLEFIA | Possible because LAT was constructible [30] | Linear cryptanalysis attack relies on LAT. | Impossible because LAT was not constructible, due to memory constraints [39] (p. 176). |
| DES | Possible because LAT was constructible [49] | Linear cryptanalysis attack relies on LAT. | Impossible because LAT was not constructible, due to memory constraints [39] (p. 176). |

**Table 9.** *Cont.*

| Algorithm | Possibility of Linear Attack before GF ($2^{32}$) Was Applied | Remarks | Possibility of Linear Attack after GF ($2^{32}$) Was Applied |
|---|---|---|---|
| MMB | Possible because LAT was constructible [50] | Linear cryptanalysis attack relies on LAT. | Impossible because LAT was not constructible, due to memory constraints [39] (p. 176). |
| RC5 | Possible because LAT was constructible [51] | Linear cryptanalysis attack relies on LAT. | Impossible because LAT was not constructible, due to memory constraints [39] (p. 176). |
| SERPENT | Possible because LAT was constructible [52] | Linear cryptanalysis attack relies on LAT. | Impossible because LAT was not constructible, due to memory constraints [39] (p. 176). |
| SKIPJACK | Possible because LAT was constructible [53] | Linear cryptanalysis attack relies on LAT. | Impossible because LAT was not constructible, due to memory constraints [39] (p. 176). |

## 6. Conclusions and Future Work

We have managed to block all procedures taken to conduct linear cryptanalysis attacks on AES, BLOWFISH, CAMELLIA, CAST, CLEFIA, DES, MMB, RC5, SERPENT, and SKIPJACK using our novel method of GF ($2^{32}$) and its long irreducible polynomial. We have proven that it is impossible to draw the LAT from a 32-bit output S-Box. We have proven that if there is no LAT, there is no linear cryptanalysis attack. Therefore, we blocked the linear cryptanalysis attack. Future work will try to block differential cryptanalysis attacks and differential–linear cryptanalysis attacks using GF ($2^{32}$) and its long irreducible polynomial. Apart from the linear cryptanalysis attack, cryptography cannot guarantee information security. Additional techniques are required to protect against attacks such as denial of service or complete system failure [24–26].

**Author Contributions:** Conceptualization, K.D.M.; methodology, K.D.M.; software, K.D.M.; validation, K.D.M. and M.S.; formal analysis, K.D.M.; investigation, K.D.M.; resources, K.D.M. and M.S.; data curation, K.D.M. and M.S.; writing—original draft preparation, K.D.M. and M.S.; writing—review and editing, K.D.M. and M.S.; visualization, K.D.M. and M.S.; supervision, M.S.; All authors have read and agreed to the published version of the manuscript.

**Funding:** This research received no external funding.

**Data Availability Statement:** The data presented in this study are available in this article.

**Conflicts of Interest:** The authors declare no conflict of interest.

## Appendix A

Figures A1 and A2 show how we use a C++ program to test and run test vectors. In the C++ code, we set temp as the key and temp2 as the plaintext. Then, we call the encryption and decryption functions; if the decryption function generates the same array as temp2, the test vectors are validated, and the algorithm is encrypting and decrypting as expected. If you compare Figures 4 and A1, the array after decryption is the same as the array of temp2. Therefore, the test vectors are valid and .dll file is the algorithm. If that was not the case, the .dll file obtained from the IoT device is not an algorithm or is corrupt.

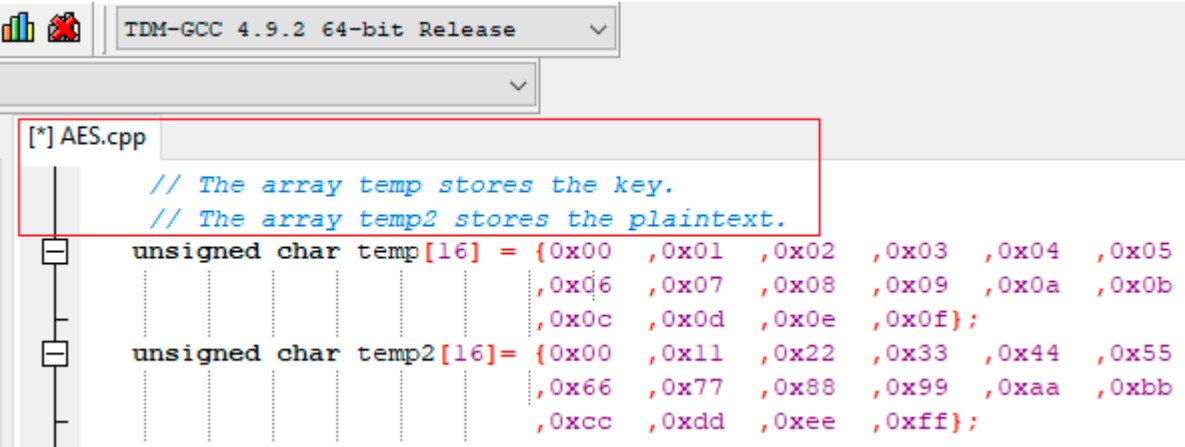

**Figure A1.** C++ coding to hard code test vectors using array.

```
Text after encryption:
69 c4 e0 d8 6a 7b 04 30 d8 cd b7 80 70 b4 c5 5a

Text after dencryption:
00 11 22 33 44 55 66 77 88 99 aa bb cc dd ee ff

-------------------------------
Process exited after 0.03851 seconds with return value 0
Press any key to continue . . . ▄
```

**Figure A2.** C++ out program of validating test vectors.

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
