# Peer review of "Blocking Linear Cryptanalysis Attacks Found on Cryptographic Algorithms Used on Internet of Thing Based on the Novel Approaches of Using Galois Field (GF (232)) and High Irreducible Polynomials"

_applsci, doi:10.3390/app132312834_

Round 1
Reviewer 1 Report
Comments and Suggestions for Authors
The paper is in part okay but many sections require a major rework before - in terms of both content as well as presentation.
1. The introduction section requires to be rewritten by including, motivations, and research contributions as well as by addressing a small outline of the paper.
2. Include literature survey earlier to your approach. This section also should need to enhance by including a summary table. Also address the limitations in the existed works.
3. The authors did not compare the complexity and performance of the proposed method with other to prove its effectiveness particularly for IoT devices as claimed in the Conclusion section.
4. How could authors can collect the algorithms directly from IoT Devices. Is it that much of easy without knowing the logic or key, etc?
5. The authors have mentioned that validate the algorithms using test vectors. Clearly explain about the test vectors.
6. In general, Linear cryptanalysis means nothing but a known plaintext attack in which the attacker studies probabilistic linear relations between parity bits of the plaintext, the ciphertext, and the secret key. So many factors have to be considered to do the attack, length, runtime, space complexity, pattern analysis, etc. Include more details about the factor considered on linear cryptanalysis attack.
7. Suggested to consider the following papers those have included various cryptanalysis.
a. https://doi.org/10.1186/s13677-023-00425-7
b. doi: 10.1109/ACCESS.2019.2917015.
c. doi: 10.1109/JBHI.2022.3178629.
Comments on the Quality of English LanguageSir/Madam,
Minor edition required for readers understanding
Author Response
Reviewer 1
The paper is in part okay but many sections require a major rework before - in terms of both content as well as presentation.
- The introduction section requires to be rewrittenby including, motivations, and research contributions as well as by addressing a small outline of the paper.
Motivations, and research contributions as well as by addressing a small outline of the paper are now include.
- Include literature survey earlier to your approach. This section also should need to enhance by including a summary table. Also address the limitations in the existed works.
Literature survey is now introduced earlier and limitation is now included.
- The authors did not compare the complexity and performance of the proposed method with other to prove its effectiveness particularly for IoT devices as claimed in the Conclusion section.
Complexity and performance are in included in the table form. Explained in the table form Table 8, 9, and 10.
- How could authors can collect the algorithms directly from IoT Devices. Is it that much of easy without knowing the logic or key, etc? i.
We collected all ten algorithms (AES, BLOWFISH, CAMELLIA, CAST, CLEFIA, DES, MMB, RC5, SERPENT, and SKIPJACK) from different IoT devices using FileDisassembler and analyzed all the .dll cryptographic files using dotPeek and iLSpy.
- The authors have mentioned that validate the algorithms using test vectors. Clearly explain about the test vectors.
Test vectors are explained in Appendix A
- In general, Linear cryptanalysis means nothing but a known plaintext attack in which the attacker studies probabilistic linear relations between parity bits of the plaintext, the ciphertext, and the secret key.So many factors have to be considered to do the attack, length, runtime, space complexity, pattern analysis, etc. Include more details about the factor considered on linear cryptanalysis attack.
Explained in the table form Table 8, 9, and 10.
- Suggested to consider the following papers those have included various cryptanalysis.
Paper were considered
Reviewer 2 Report
Comments and Suggestions for Authors
Congratulations to the authors on a very interesting and useful work. My only wish is to discuss the possibilities of the approach for more complex cases of cryptographic encryption, of which differential encryption seems to be the most interesting for the Internet of Things.
Author Response
Reviewer 2
Congratulations to the authors on a very interesting and useful work. My only wish is to discuss the possibilities of the approach for more complex cases of cryptographic encryption, of which differential encryption seems to be the most interesting for the Internet of Things.
Explained in the table form Table 8, 9, and 10.
Reviewer 3
To block linear cryptanalysis attack, the authors applied a mathematical novel approach called Galois Field (GF(232)) and high irreducible polynomials to be mapped on the S-Box. Results are correct and interesting. However, to improve the quality and presentation of the paper, the authors are suggested to address the following comments
- In Fig. 2, the author depicts the flowchart for obtaining the key, and I think the whole process of obtaining the key by the encryption algorithm should be analyzed theoretically.
We collected all ten algorithms (AES, BLOWFISH, CAMELLIA, CAST, CLEFIA, DES, MMB, RC5, SERPENT, and SKIPJACK) from different IoT devices using FileDisassembler and analyzed all the .dll cryptographic files using dotPeek and iLSpy.
- Please carefully check and modify whether the labels in the figure are written in a standardized manner, as well as the sorting and citation of references.
All are now checked
- In this paper, the authors applied a mathematical method called Galois Field to stop linear cryptanalysis attacks. I think some references on cryptographic application should be considered to compare their advantage, such as, Exploiting flexible and secure cryptographic technique for multi-dimensional image based on graph data structure and three-input majority gate. A memristive fully connect neural network and application of medical image encryption based on central diffusion algorithm.
Explained in the table form Table 8, 9, and 10.
- In the AES encryption algorithm in Chapter 9, the authors mapped the GF and its long irreducible polynomials to obtain a new S-Box with 32-bit output. please briefly describe the process of generating S-Box and provide examples for explanation.
This is explained in Section 3, Table 6 and in Table 7
- The disadvantages of the proposed method may be given in the conclusion.
Disadvantage of the proposed method is give in the conclusion and even in section 2.1
Reviewer 3 Report
Comments and Suggestions for Authors
To block linear cryptanalysis attack, the authors applied a mathematical novel approach called Galois Field (GF(232)) and high irreducible polynomials to be mapped on the S-Box. Results are correct and interesting. However, to improve the quality and presentation of the paper, the authors are suggested to address the following comments
1. In Fig. 2, the author depicts the flowchart for obtaining the key, and I think the whole process of obtaining the key by the encryption algorithm should be analyzed theoretically.
2. Please carefully check and modify whether the labels in the figure are written in a standardized manner, as well as the sorting and citation of references.
3. In this paper, the authors applied a mathematical method called Galois Field to stop linear cryptanalysis attacks. I think some references on cryptographic application should be considered to compare their advantage, such as, Exploiting flexible and secure cryptographic technique for multi-dimensional image based on graph data structure and three-input majority gate. A memristive fully connect neural network and application of medical image encryption based on central diffusion algorithm.
4. In the AES encryption algorithm in Chapter 9, the authors mapped the GF and its long irreducible polynomials to obtain a new S-Box with 32-bit output. please briefly describe the process of generating S-Box and provide examples for explanation.
5. The disadvantages of the proposed method may be given in the conclusion.
Comments on the Quality of English Language
Moderate editing of English language required
Author Response

(The authors gave the same response as above.)

Round 2
Reviewer 1 Report
Comments and Suggestions for Authors
No objections